# How Was the Staple Food Supply Chain in Indonesia Affected by COVID-19?

Eka Purna Yudha [1,*] and Julian Roche [2]

1  Department of Agricultural Socio-Economics, Universitas Padjadjaran, Sumedang 45363, Indonesia
2  School of Agricultural and Resource Economics, University of Western Australia, Perth 6009, Australia; julian.roche@uwa.edu.au
*  Correspondence: eka.purna.yudha@unpad.ac.id

**Abstract:** During the COVID-19 pandemic, there were significant restrictions on the transportation of food products in Indonesia. The research objective of this study was to investigate the extent to which these restrictions impacted changes in marketing margins at the provincial level in Indonesia. The approach taken was through the examination of trade and freight margin statistical data before the pandemic (2019) and after the pandemic (2020) across a number of different commodity markets: rice, shallots, red chilli pepper, beef, chicken meat and eggs, sugar, and cooking oil. The evidence indicates that the pandemic brought a rapid rise in Indonesian domestic prices as a result of purchasing panic at its start. But after the imposition of transportation restrictions, there were wide variations: some durable food options experienced increased marketing margins, whereas non-durables tended to experience decreased marketing margins in some regions, as fresh products such as red chillies and shallots were discarded as a result of declining consumer purchasing power. The conclusion for policymakers is that any future restrictions should take into account this likely difference in response, in order to minimise economic disruption by calibrating support along the supply chain.

**Keywords:** marketing margins; supply chains; COVID-19 pandemic; staple food

## 1. Introduction

Global food supply networks were hampered by the COVID-19 pandemic (Béné 2020). The disruptions in the flow of products and services, especially staple foods, have been attributed to lockdowns, border restrictions, labour shortages, and logistical difficulties. For disadvantaged people in particular, these disturbances had a cascading impact on food access, affordability, and nutrition (Debucquet et al. 2020). Understanding the scope of these disruptions is essential for developing successful policies and interventions in Indonesia since the majority of the population depends on readily accessible staples. The importance of ensuring food supplies during a pandemic is obvious, especially in developing countries where supply chains are regularly stressed (Narula 2017, p. 214), and where supply chain stress tests have now been recommended after the pandemic (Sharma et al. 2021), but there is little empirical research into how COVID-19 itself impacted supply chains (Barman et al. 2021). The research question addressed in this paper is in what ways the agribusiness supply chain in a specific region of Indonesia was affected by COVID-19.

Movement restrictions and health protocols that were used to identify the virus disrupted daily activities. Many farmers experience stress when managing jobs, such as migrant work, because of the stress of the journey. It can affect the processes of planting, tending, and harvesting. In addition to that, the deterioration of fertilisers, seeds, and other materials reduces productivity. Employees who are ill or who must perform quarantine procedures may also disrupt the production line's output. This is related to the prevalence of processed products and their prompt consumption on the street. Supply logistics is harmed by border movements and border closures.

Reducing the movement of people and reducing the capacity of air transportation causes difficulties in sending agricultural products from one region to another, both domestically and internationally. This may prevent price increases and decreases in the availability of certain goods in a number of locations. Additionally, pandemic-long consumer behaviour also worsens food supply chains. Many restaurants and coffee shops offer Tutup Service or accept deliveries through shipping services, which can satisfy requests for specific food items. Some food types may experience a spike in demand, while others may have a drastic decline. Fluctuations in the price of staple food may be caused by disturbances in the supply chain. Decreasing production or availability may prevent the price from rising, especially if demand remains high. This may be detrimental to the economy and the capacity of the population to meet basic needs. This government has acknowledged the significance of having a flexible and adaptable food system for handling crises. Numerous nations feel the need to increase domestic food production and minimise trade costs.

As an archipelagic country with a very wide area, inadequate logistics infrastructure, and high transportation costs, Indonesia poses serious challenges for the supply and distribution of agricultural commodities and basic commodities. It is impossible to overestimate the significance of Indonesia's supply chains for basic foods. It is the fourth most populated nation in the world, with a population of more than 270 million. This situation requires synchronisation and the alignment of progress between economic sectors and between regions for the realisation of better and more equitable economic growth (Dermoredjo et al. 2021). In addition, in improving the economy of a region, attention to logistics distribution channels has a very strategic role in the midst of the COVID-19 pandemic, especially those related to food. The flow of food trade can be carried out by land, sea, and air, so each of these routes becomes a concern in terms of the distribution of these commodities throughout Indonesia (Hirawan and Verselita 2020). The stability of food supply chains is a crucial national concern given this demographic size. Disruptions in these supply chains can result in both short-term food shortages and long-term socioeconomic effects in a nation that predominantly depends on domestically produced staples to meet the nutritional demands of its inhabitants (Aryani et al. 2015).

The economic sector that was worst hit during this pandemic was transportation and warehousing services, which contracted by $-30.84\%$ in the second quarter of 2020 (year on year). Almost all business fields in this sector shared in this contraction, with rail, land, sea, river, air, and warehousing transportation falling by $-63.75$, $-17.65$, $-17.48$, $-26.66$, $-80.23$, and $-38.69\%$, respectively (BPS 2023). The slowdown due to the COVID-19 pandemic had a significant impact on the availability of domestic agricultural and food commodities due to the disruption of the national logistics system.

The Central Bureau of Statistics of Indonesia uses eight commodities in conducting its annual survey of distribution patterns. These commodities include medium rice, shallots (red onions), red chillies, beef, purebred chicken, chicken eggs, granulated sugar, and cooking oil. We have been able to analyse the trade and transport margin at the level of individual Indonesian provinces using an innovative-paradigm, providing fresh perspectives by offering in-depth insights that significantly extend beyond what was previously available. By integrating methodology with regional spatial analysis, our research tracks market evolution and provides a solid foundation for articulating strategic goals under current circumstances. This study will focus on the impact shock of the COVID-19 pandemic on the distribution supply chain in each province on the eight main agricultural commodities in domestic trade.

## 2. Methodology

Eight commodities/foodstuffs were selected for this study, based on their consumption and contribution to inflation, as identified by the National Strategic Food Price Information Center and as employed in previous studies (Laksono and Yuliawati 2021; Laili et al. 2022; Nurdjannah et al. 2014). Table 1 provides information on each. Data analysis was carried

out through the use of descriptive statistical methods by calculating the difference between the differences in trade and freight margin data before the pandemic (2019) and after the pandemic (2020). The results of these calculations are spatially illustrated on a provincial scale in Indonesia.

**Table 1.** Characteristics of agricultural commodities selected in this study.

| Commodity | Product Characteristics | General Pattern of Distribution | Market Characteristics |
|---|---|---|---|
| Medium rice | It can be stored for years (Pahulu et al. 2007). | Manufacturers --> distributors (1 . . . n) --> retailers --> final consumers | At the initial level (farmers to rice mills) the market is oligopsony, and after that, the market is more oligopoly (InterCAFE and LPPM 2018). |
| Shallots | It can be stored for about 1 month indoors (Tesfa et al. 2015). | Farmers --> collectors (1 . . . n) --> retailers --> final consumers | From the market at the farmer level to wholesalers, the market has an oligopsony structure but becomes an oligopoly at the wholesale level to the retail level (InterCAFE and LPPM 2018). |
| Red chilli pepper | Products are easily damaged from harvest, sorting, and storage until they are in the hands of consumers (Nurdjannah et al. 2014). | Farmers --> wholesalers (1 . . . n) --> retailers --> final consumers | From farmers to wholesalers, the market is oligopsony, and from wholesalers to retail levels, it has an oligopoly structure (InterCAFE and LPPM 2018; (Sativa et al. 2017). |
| Beef | It can be stored in the freezer at a temperature below −18 degrees Celsius for 6–12 months. | Producer line: producer --> wholesaler --> retailer --> final consumer Importer line: importer --> distributor (1 . . . n) -- -> retailer --> final consumer | It is a tight oligopoly market (Setiaji et al. 2017). |
| Chicken meat | Whole raw chicken meat that is frozen, will last up to one year. | Producers --> collectors (1 . . . n) --> retailers --> final consumers | An oligopsonistic market structure where prices are determined more by collectors/brokers (Saptana et al. 2015). |
| Chicken eggs | Eggs can last up to 4 to 5 weeks if stored in the refrigerator. On the other hand, when stored indoors, eggs will rot faster and can only last up to 3 weeks. | Producers --> collectors (1 . . . n) --> retailers --> final consumers | It has an oligopoly market and perfect competition due to the homogeneous nature and characteristics of the commodity and it is a mass commodity (Octaviani et al. 2013). |
| Sugar | Products from the sugarcane raw material processing industry, and have a relatively long shelf life. | Manufacturer --> distributor --> wholesaler --> retailer --> final consumer | The sale of sugar in the early stages of the marketing chain is carried out using an auction system and is controlled by SOEs. Along the market chain, it was found that seepage of refined crystal sugar (imported sugar) into the local sugar market was found (InterCAFE and LPPM 2018). |
| Cooking oil | Palm oil processing industry products have a relatively long shelf life. | Manufacturers --> distributors --> retailers --> end consumers | The market structure is oligopsony in the early stages and is structured as oligopoly in the later stages (InterCAFE and LPPM 2018). |

We then used secondary data obtained from the central statistical agency of Indonesia in 2019 and 2020. The data used are trade and transport margin (MPP) data at the provincial level in Indonesia, shown below in Table 2. Trade and freight margin (MPP) is the compensation of traders as distributors of goods, which is the difference between the sales value and the purchase value. The study limitations of this research pertain to trade and

transport margin data for the provincial level in Indonesia by using staple data specific to the country of Indonesia. The MPP margin is a measure of the size of the output from trading activities (BPS 2023).

Mathematically, it can be written as follows:

$$M_{ji} = P_{si} - P_{bi}, \text{ or} \tag{1}$$

$$M_{ji} = b_{ti} - \pi_i, \text{ or} \tag{2}$$

$$\pi_i = M_{ji} - b_{ti} \tag{3}$$

The total marketing margin (M) can be mathematically written as follows:

$$M_j = \sum_{i=1}^{n} M_{ji}, \text{ or } M_j = P_r - P_f \tag{4}$$

Information:

$M_{ji}$ = i-level marketing agency margins;
$P_{si}$ = sales price i-level marketing agency;
$P_{bi}$ = purchase price i-level marketing agency;
$b_{ti}$ = marketing fee i-level marketing agency;
$\pi_i$ = advantages of i-level marketing agency;
$M_j$ = total marketing margin;
$P_r$ = price at the consumer level;
$P_f$ = price at the producer level.

**Table 2.** Trade and transport margin data (MPP) of the main agricultural commodities for each province in Indonesia (parts 1 and 2).

| NO | Province | Rice | | | Sugar | | | Cooking Oil | | | Chicken Eggs | | |
|---|---|---|---|---|---|---|---|---|---|---|---|---|---|
| | | 2019 | 2020 | Change | 2019 | 2020 | Change | 2019 | 2020 | Change | 2019 | 2020 | Change |
| 1 | Aceh | 5.91 | 13.1 | 7.19 | 17.63 | 37.43 | 19.8 | 22.7 | 18 | −4.7 | 12.03 | 15.56 | 3.53 |
| 2 | Sumatera Utara | 20.97 | 15.13 | −5.84 | 12.31 | 18.35 | 6.04 | 21.98 | 12.13 | −9.85 | 23.98 | 20.32 | −3.66 |
| 3 | Sumatera Barat | 12.99 | 15.3 | 2.31 | 18.47 | 37.83 | 19.36 | 26.77 | 10.43 | −16.34 | 38.51 | 19.57 | −18.94 |
| 4 | Riau | 18.14 | 20.97 | 2.83 | 27.55 | 17.18 | −10.37 | 29.01 | 22.03 | −6.98 | 52.87 | 16.45 | −36.42 |
| 5 | Jambi | 23.12 | 9.53 | −13.59 | 26.88 | 18.24 | −8.64 | 16.97 | 17.39 | 0.42 | 9.62 | 9.05 | −0.57 |
| 6 | Sumatera Selatan | 11.68 | 20.91 | 9.23 | 9.7 | 19.42 | 9.72 | 18.82 | 25.08 | 6.26 | 18.44 | 12.34 | −6.1 |
| 7 | Bengkulu | 4.97 | 13.49 | 8.52 | 27.52 | 18.22 | −9.3 | 28.36 | 21.95 | −6.41 | 33.93 | 5.5 | −28.43 |
| 8 | Lampung | 7.13 | 14.45 | 7.32 | 20.29 | 12.53 | −7.76 | 22.86 | 13.06 | −9.8 | 23.45 | 19.34 | −4.11 |
| 9 | Kep. Bangka Belitung | 22.74 | 20.04 | −2.7 | 23.13 | 19.16 | −3.97 | 26.22 | 14.3 | −11.92 | 24.62 | 8.32 | −16.3 |
| 10 | Kep. Riau | 29.03 | 27.12 | −1.91 | 40.68 | 31.06 | −9.62 | 33.48 | 34.18 | 0.7 | 23.69 | 6.39 | −17.3 |
| 11 | Dki Jakarta | 37.67 | 20.01 | −17.66 | 25.35 | 31.82 | 6.47 | 14.73 | 17.62 | 2.89 | 21.41 | 18.36 | −3.05 |
| 12 | Jawa Barat | 10.64 | 17.86 | 7.22 | 22.99 | 30.6 | 7.61 | 35.1 | 21.64 | −13.46 | 22.63 | 22.34 | −0.29 |
| 13 | Jawa Tengah | 9.32 | 19.2 | 9.88 | 10.79 | 15.24 | 4.45 | 14.68 | 15.77 | 1.09 | 10.91 | 16.1 | 5.19 |
| 14 | Di Yogyakarta | 14.82 | 17.78 | 2.96 | 19.38 | 12.02 | −7.36 | 15.55 | 13.96 | −1.59 | 37.55 | 10.72 | −26.83 |
| 15 | Jawa Timur | 34.15 | 19.22 | −14.93 | 19.3 | 18.09 | −1.21 | 25.59 | 16.39 | −9.2 | 10.21 | 16.17 | 5.96 |
| 16 | Banten | 12.42 | 9.12 | −3.3 | 34.83 | 15.49 | −19.34 | 37.34 | 19.06 | −18.28 | 21.07 | 19.92 | −1.15 |
| 17 | Bali | 16.06 | 15.74 | −0.32 | 25.45 | 17.86 | −7.59 | 35.6 | 17.65 | −17.95 | 36.33 | 20.44 | −15.89 |
| 18 | Nusa Tenggara Barat | 4.01 | 15.14 | 11.13 | 15.5 | 30.62 | 15.12 | 7.91 | 15.2 | 7.29 | 25.05 | 25.05 | 0 |
| 19 | Nusa Tenggara Timur | 23.51 | 11.85 | −11.66 | 25.51 | 45.45 | 19.94 | 18.71 | 26.31 | 7.6 | 21.89 | 38.94 | 17.05 |
| 20 | Kalimantan Barat | 14.17 | 11.22 | −2.95 | 19.01 | 45.77 | 26.76 | 30.05 | 26.3 | −3.75 | 20.02 | 12.5 | −7.52 |
| 21 | Kalimantan Tengah | 14.21 | 17.06 | 2.85 | 31.86 | 16.15 | −15.71 | 12.69 | 23.67 | 10.98 | 28.92 | 15.78 | −13.14 |
| 22 | Kalimantan Selatan | 13.63 | 11.99 | −1.64 | 19.07 | 16.33 | −2.74 | 30.13 | 18.47 | −11.66 | 25.91 | 17.59 | −8.32 |
| 23 | Kalimantan Timur | 10.76 | 11.12 | 0.36 | 20.61 | 17.32 | −3.29 | 13.9 | 23.77 | 9.87 | 31.31 | 16.31 | −15 |
| 24 | Kalimantan Utara | 24.26 | 19.77 | −4.49 | 16.27 | 17.34 | 1.07 | 27.85 | 20.33 | −7.52 | 11.59 | 27.71 | 16.12 |
| 25 | Sulawesi Utara | 14.52 | 18.98 | 4.46 | 21.46 | 45.12 | 23.66 | 22.93 | 28.29 | 5.36 | 7.07 | 19.15 | 12.08 |
| 26 | Sulawesi Tengah | 8.69 | 6.09 | −2.6 | 20.44 | 36.41 | 15.97 | 43.86 | 27.57 | −16.29 | 23.87 | 18.74 | −5.13 |
| 27 | Sulawesi Selatan | 21.62 | 18.62 | −3 | 16.53 | 24.85 | 8.32 | 20.5 | 24.65 | 4.15 | 12.76 | 14.56 | 1.8 |
| 28 | Sulawesi Tenggara | 9.51 | 12.13 | 2.62 | 40.26 | 29.49 | −10.77 | 43.83 | 31.33 | −12.5 | 13.87 | 19.9 | 6.03 |

**Table 2.** *Cont.*

| NO | Province | Rice | | | Sugar | | | Cooking Oil | | | Chicken Eggs | | |
|---|---|---|---|---|---|---|---|---|---|---|---|---|---|
| | | 2019 | 2020 | Change | 2019 | 2020 | Change | 2019 | 2020 | Change | 2019 | 2020 | Change |
| 29 | Gorontalo | 18.17 | 18.87 | 0.7 | 15.05 | 34.3 | 19.25 | 33.24 | 25.42 | −7.82 | 17.05 | 23.27 | 6.22 |
| 30 | Sulawesi Barat | 22.23 | 15.82 | −6.41 | 11.12 | 25.52 | 14.4 | 20.65 | 22.85 | 2.2 | 28.16 | 14.86 | −13.3 |
| 31 | Maluku | 32.71 | 26.47 | −6.24 | 57.49 | 31.5 | −25.99 | 42.95 | 18.39 | −24.56 | 32.37 | 42.99 | 10.62 |
| 32 | Maluku Utara | 19.03 | 25.4 | 6.37 | 43.58 | 23.4 | −20.18 | 29.91 | 30.86 | 0.95 | 9.69 | 26.69 | 17 |
| 33 | Papua Barat | 19.91 | 24.75 | 4.84 | 40.46 | 29.62 | −10.84 | 24.48 | 33.92 | 9.44 | 25.31 | 30.01 | 4.7 |
| 34 | Papua | 22.23 | 25.13 | 2.9 | 31.44 | 36.5 | 5.06 | 39.76 | 37.26 | −2.5 | 15 | 18.13 | 3.13 |
| 35 | Indonesia | 22.34 | 21.47 | −0.87 | 33.18 | 25.86 | −7.32 | 17.05 | 17.41 | 0.36 | 13.09 | 20.19 | 7.1 |
| NO | Province | Beef | | | Shallot | | | Red Chilli Pepper | | | Chicken Beef | | |
| | | 2019 | 2020 | Change | 2019 | 2020 | Change | 2019 | 2020 | Change | 2019 | 2020 | Change |
| 1 | Aceh | 14.18 | 7.24 | −6.94 | 31.29 | 69.59 | 38.3 | 53.66 | 83.99 | 30.33 | 25.23 | 45.52 | 20.29 |
| 2 | Sumatera Utara | 18.37 | 21.69 | 3.32 | 50.17 | 46.02 | −4.15 | 26.19 | 47.75 | 21.56 | 23.04 | 20.5 | −2.54 |
| 3 | Sumatera Barat | 18.01 | 16.05 | −1.96 | 17.46 | 41.06 | 23.6 | 61.84 | 49.98 | −11.86 | 33.59 | 34.95 | 1.36 |
| 4 | Riau | 11.97 | 6.53 | −5.44 | 36.42 | 39.77 | 3.35 | 49.11 | 57.14 | 8.03 | 43.52 | 23.46 | −20.06 |
| 5 | Jambi | 8.72 | 20.48 | 11.76 | 46.33 | 16.34 | −29.99 | 52.29 | 48.28 | −4.01 | 4.89 | 39.13 | 34.24 |
| 6 | Sumatera Selatan | 24.44 | 35.75 | 11.31 | 38.9 | 36.27 | −2.63 | 56.74 | 61.06 | 4.32 | 19.46 | 17.6 | −1.86 |
| 7 | Bengkulu | 56.54 | 65.34 | 8.8 | 41.16 | 37.46 | −3.7 | 81.44 | 45.42 | −36.02 | 80.2 | 63.76 | −16.44 |
| 8 | Lampung | 51.5 | 24.41 | −27.09 | 60.95 | 34.53 | −26.42 | 18.3 | 31.1 | 12.8 | 61.18 | 58.89 | −2.29 |
| 9 | Kep. Bangka Belitung | 26.28 | 47.77 | 21.49 | 29.17 | 60.43 | 31.26 | 66.26 | 57.43 | −8.83 | 26.99 | 36.75 | 9.76 |
| 10 | Kep. Riau | 20.7 | 22.98 | 2.28 | 35.17 | 26.81 | −8.36 | 42.21 | 71.41 | 29.2 | 73.2 | 49.59 | −23.61 |
| 11 | Dki Jakarta | 23.4 | 41.71 | 18.31 | 26.82 | 41.14 | 14.32 | 25.69 | 77.84 | 52.15 | 19.23 | 43.55 | 24.32 |
| 12 | Jawa Barat | 15.78 | 23.4 | 7.62 | 46.04 | 31.37 | −14.67 | 82.31 | 80.73 | −1.58 | 24.51 | 24.77 | 0.26 |
| 13 | Jawa Tengah | 13.77 | 16.86 | 3.09 | 44.79 | 23.85 | −20.94 | 61.01 | 34.25 | −26.76 | 33.67 | 21.57 | −12.1 |
| 14 | Di Yogyakarta | 22.63 | 20.25 | −2.38 | 60.53 | 27.97 | −32.56 | 46.72 | 42.48 | −4.24 | 22.06 | 33.37 | 11.31 |
| 15 | Jawa Timur | 16.05 | 27.96 | 11.91 | 39.76 | 100.57 | 60.81 | 56.52 | 56.09 | −0.43 | 56.42 | 42.11 | −14.31 |
| 16 | Banten | 15.4 | 23.02 | 7.62 | 48.94 | 28.35 | −20.59 | 27.8 | 38.56 | 10.76 | 40.76 | 58.99 | 18.23 |
| 17 | Bali | 14.8 | 29.26 | 14.46 | 33.09 | 25.68 | −7.41 | 41.82 | 16.53 | −25.29 | 32.33 | 23.02 | −9.31 |
| 18 | Nusa Tenggara Barat | 7.94 | 16.3 | 8.36 | 80.83 | 22.85 | −57.98 | 46.48 | 35.31 | −11.17 | 38.5 | 29.16 | −9.34 |
| 19 | Nusa Tenggara Timur | 15.3 | 25.03 | 9.73 | 27.27 | 63.3 | 36.03 | 33.89 | 52.72 | 18.83 | 43.72 | 38.63 | −5.09 |
| 20 | Kalimantan Barat | 21.36 | 36.08 | 14.72 | 62.46 | 50.4 | −12.06 | 57.9 | 47.9 | −10 | 57.44 | 29.37 | −28.07 |
| 21 | Kalimantan Tengah | 21.92 | 47.02 | 25.1 | 48.78 | 57.47 | 8.69 | 106.21 | 98.69 | −7.52 | 17.22 | 21.27 | 4.05 |
| 22 | Kalimantan Selatan | 19.87 | 6.8 | −13.07 | 59.41 | 44.68 | −14.73 | 56.7 | 46.62 | −10.08 | 17.28 | 17.91 | 0.63 |
| 23 | Kalimantan Timur | 14.22 | 49.9 | 35.68 | 41.2 | 31.65 | −9.55 | 50.04 | 77.21 | 27.17 | 57.87 | 40.3 | −17.57 |
| 24 | Kalimantan Utara | 20 | 38.39 | 18.39 | 87.8 | 108.44 | 20.64 | 61.95 | 63.11 | 1.16 | 64.98 | 82.96 | 17.98 |
| 25 | Sulawesi Utara | 16.57 | 23.12 | 6.55 | 39.89 | 83.49 | 43.6 | 11.01 | 69.02 | 58.01 | 73.88 | 60.28 | −13.6 |
| 26 | Sulawesi Tengah | 11.54 | 40.26 | 28.72 | 60 | 88.9 | 28.9 | 91.86 | 27.78 | −64.08 | 37.27 | 29.8 | −7.47 |
| 27 | Sulawesi Selatan | 12.04 | 9.4 | −2.64 | 60.62 | 29.44 | −31.18 | 64.76 | 43.89 | −20.87 | 35.2 | 25.3 | −9.9 |
| 28 | Sulawesi Tenggara | 17.37 | 13.72 | −3.65 | 67.48 | 124.9 | 57.42 | 61.96 | 88.26 | 26.3 | 37.05 | 36.75 | −0.3 |
| 29 | Gorontalo | 13.55 | 50.29 | 36.74 | 55.49 | 98.82 | 43.33 | 38.09 | 47.48 | 9.39 | 22.22 | 43.4 | 21.18 |
| 30 | Sulawesi Barat | 21.99 | 31.42 | 9.43 | 53.25 | 61.16 | 7.91 | 44.44 | 28.77 | −15.67 | 43.2 | 49.87 | 6.67 |
| 31 | Maluku | 16.21 | 17.12 | 0.91 | 78.06 | 62.72 | −15.34 | 26.27 | 29.9 | 3.63 | 27.43 | 60.03 | 32.6 |
| 32 | Maluku Utara | 10.95 | 21.15 | 10.2 | 59.63 | 105.91 | 46.28 | 98.52 | 81.31 | −17.21 | 55.63 | 30.42 | −25.21 |
| 33 | Papua Barat | 37.27 | 11 | −26.27 | 86.44 | 134.78 | 48.34 | 110.91 | 41.09 | −69.82 | 68.87 | 34.55 | −34.32 |
| 34 | Papua | 19.46 | 24.52 | 5.06 | 68.14 | 30.09 | −38.05 | 56.08 | 34.47 | −21.61 | 55.92 | 36.38 | −19.54 |
| 35 | Indonesia | 34.11 | 41.04 | 6.93 | 35.73 | 38.01 | 2.28 | 43.09 | 61.31 | 18.22 | 24.72 | 25.53 | 0.81 |

Source: 2022 analysis results.

## 3. Results

The COVID-19 pandemic had an impact on the supply of key food commodities. This happened as a whole as a result of the policy of restricting population mobility, which affected the availability of food in the consumer market, so that prices in consumer and producer markets experienced different changes between regions (Poudel et al. 2020; Siche 2020). The results of our data analysis focus on the changes in the pattern of marketing margins for medium rice, shallots, red chillies, beef, purebred chicken, eggs, granulated sugar, and cooking oil in each province due to the emergence of the COVID-19 pandemic.

### 3.1. The Margin Change Pattern for Medium Rice Market

Rice is a strategic staple commodity supervised by the government and with its distribution intervention also carried out by the government. Price variations in the rice market were relatively lower than price variations in other food commodity markets, either before or during the COVID-19 pandemic (Dermoredjo et al. 2021). The relatively more

stable price for the rice market cannot be separated from the relatively more intensive government intervention compared with other food markets (Herawati and Harianto 2021).

Most of the production centres for food commodities are located in Java and South Sulawesi. Based on Figure 1, it can be seen that the COVID-19 pandemic only had an effect on increasing the marketing margin of medium rice commodities in several provinces such as West Java, Central Java, most provinces on Sumatra Island, East Kalimantan, Central Kalimantan, a small number of provinces on Sulawesi Island, North Maluku, Papua, and West Papua. The mobility restriction policy in March–July 2020 also affected the rice distribution chain system, thus triggering an increase in prices originating in several provinces that have high consumption needs.

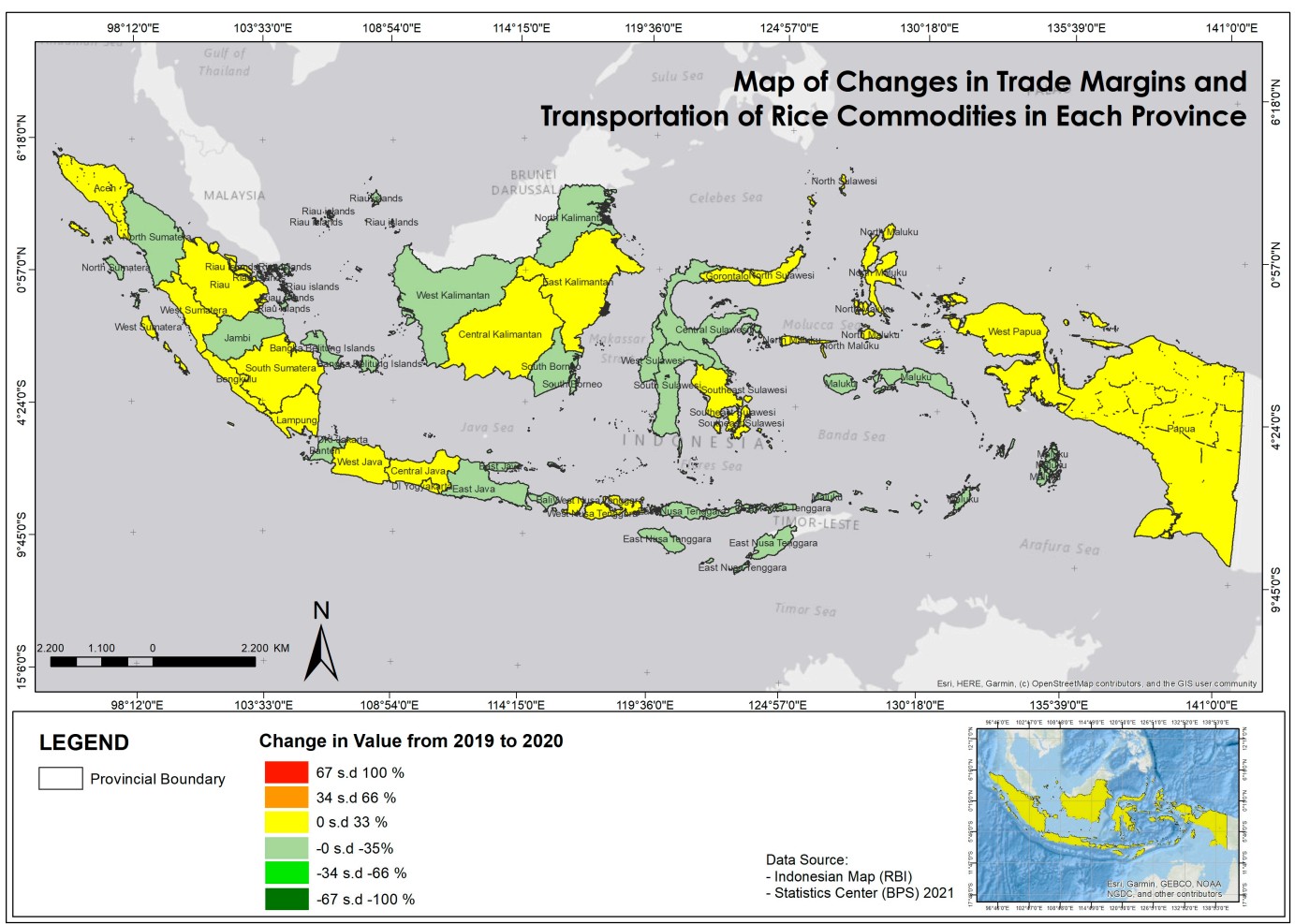

**Figure 1.** Map of margin changes for medium rice commodity marketing at the provincial level (source: Indonesian Map and Statistics Center 2021, analysed).

### 3.2. The Margin Change Pattern for Shallot Market

Shallots are included in the ten staple ingredients that have strategic value both politically and economically (Laksono and Yuliawati 2021; Amanda et al. 2016). Shallots in Indonesia are staple commodities that are difficult to replace, so regardless of the price of shallots, the demand will always be directly proportional to the high level of consumption. The price movement of shallots in each market has high fluctuations with a downward trend, as shown in Figure 2. The provinces of Central Java and West Java are the centres of shallot production in Indonesia. However, some provinces that have access to marketing chains were far from experiencing a fairly high increase in marketing margins during the pandemic. These provinces include Aceh, East Java, Southeast Sulawesi, North Sulawesi, East Nusa Tenggara, North Maluku, and West Papua.

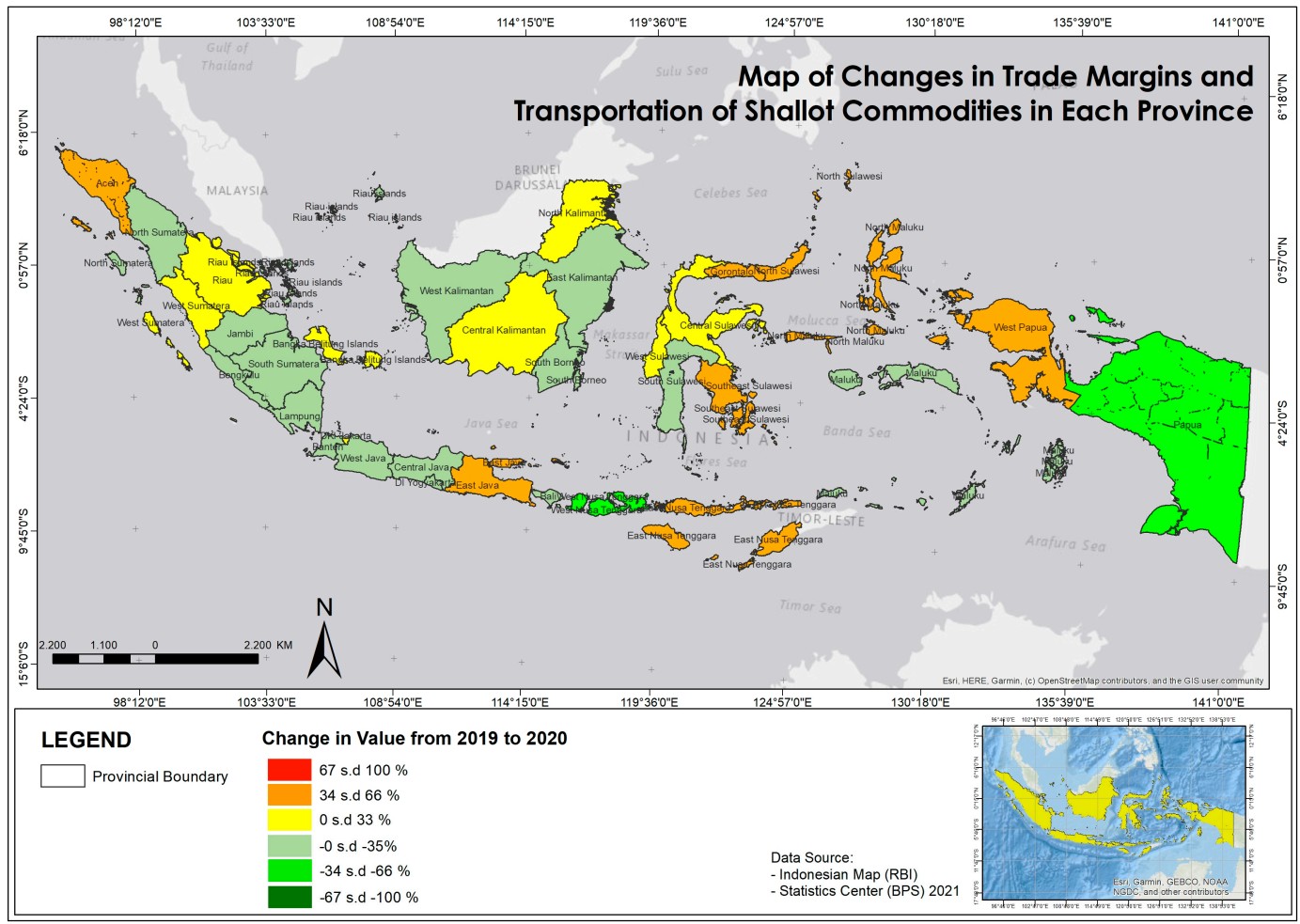

**Figure 2.** Map of changes in marketing margin of shallots at the provincial level (source: Indonesian Map and Statistics Center 2021, analysed).

### 3.3. The Margin Change Pattern for the Red Chilli Market

The marketing margin in the red chilli commodity market, which is perishable, is relatively larger than the marketing margin for non-perishable foodstuffs such as rice, cooking oil, and granulated sugar (Varshney et al. 2020). Marketing margins not only increased in magnitude in the period following the COVID-19 pandemic but also showed increased variability, as shown in Figure 3. However, interesting findings are obtained in the case of Indonesia, where there are variations in marketing margin changes for red chilli commodities at the provincial level. Provinces that have a characteristic consumption pattern in which red chillies are used more in cooking menus experience higher marketing margins. This pattern of increasing marketing margins is widely spread in most provinces on Sumatra Island, East–North Kalimantan Island, North Sulawesi, East Nusa Tenggara, and Maluku.

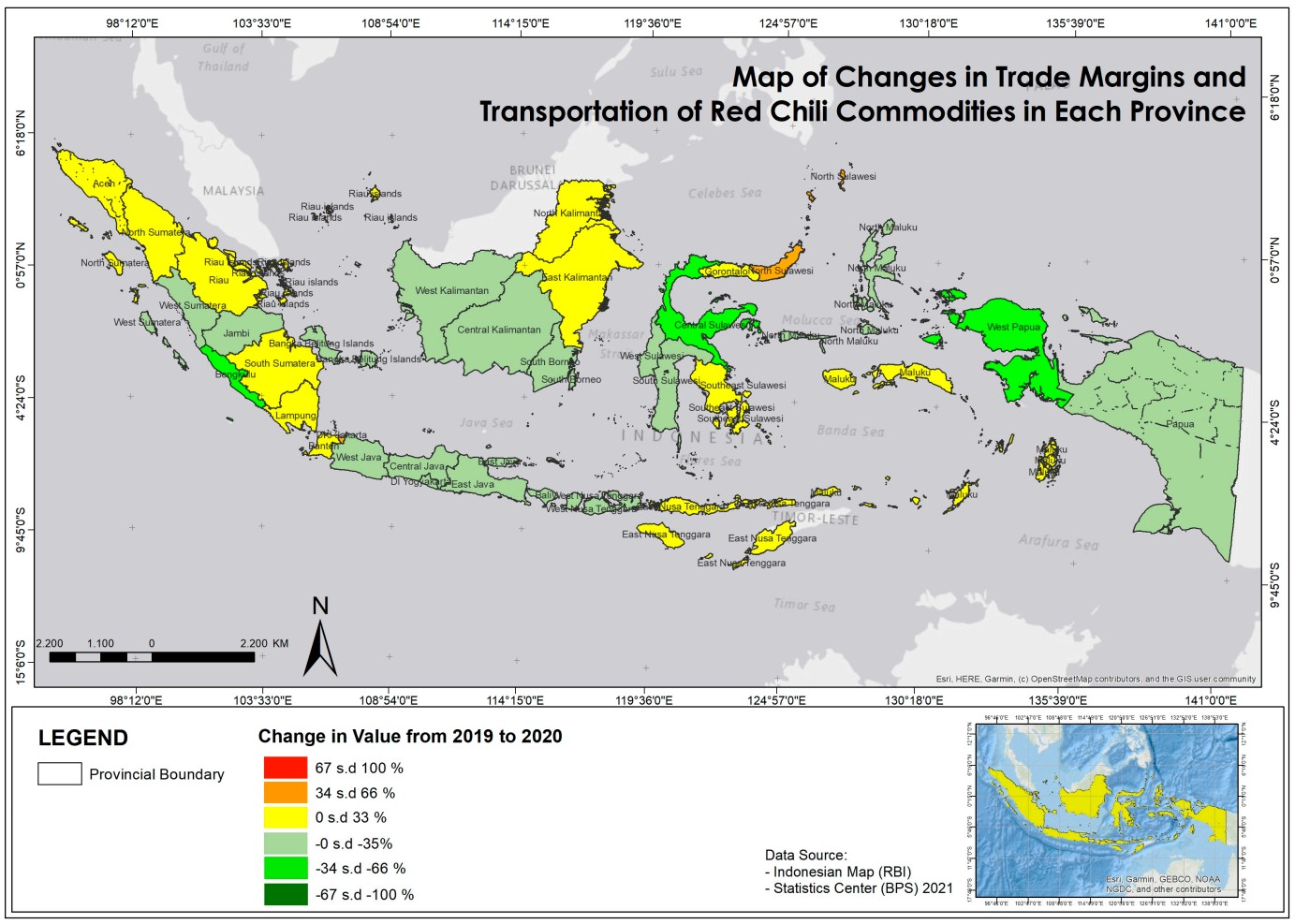

**Figure 3.** Map of margin changes for the red chilli commodity market at the provincial level (source: Indonesian Map and Statistics Center 2021, analysed).

### 3.4. The Margin Change Pattern in the Beef Commodity Market

Beef was the commodity with the most affected marketing margin due to the pandemic throughout 2020. A total of 23 provinces experienced an increase in their beef marketing margin, see Figure 4 below. Provinces located on the islands of Java, Bali, and Nusa Tenggara were the areas that experienced a significant increase in marketing margins. However, East Kalimantan and Gorontalo were the provinces that experienced the highest margin increase, namely 36.68 percent and 36.74 percent, respectively.

This increase in margin was influenced by various factors such as policies imposing restrictions on mobility, which caused supply distribution to slow down; panic buying of high protein products among the middle- and upper-class economies for household stock purposes; and decreased production capacity due to a general decline in demand for beef due to increased layoffs in most factories in Indonesia. According to a cattle and beef distributor in Bekasi, national beef sales turnover during the COVID-19 pandemic was only 60% of pre-pandemic levels; there were undoubtedly delays caused by large-scale social restrictions, low implementation of sanitation and hygiene, physical distancing, and a lack of discipline in the use of personal equipment (Zulmaneri et al. 2021). This increase in marketing margins may also indicate that demand for beef is more responsive than staple food commodities in Indonesia.

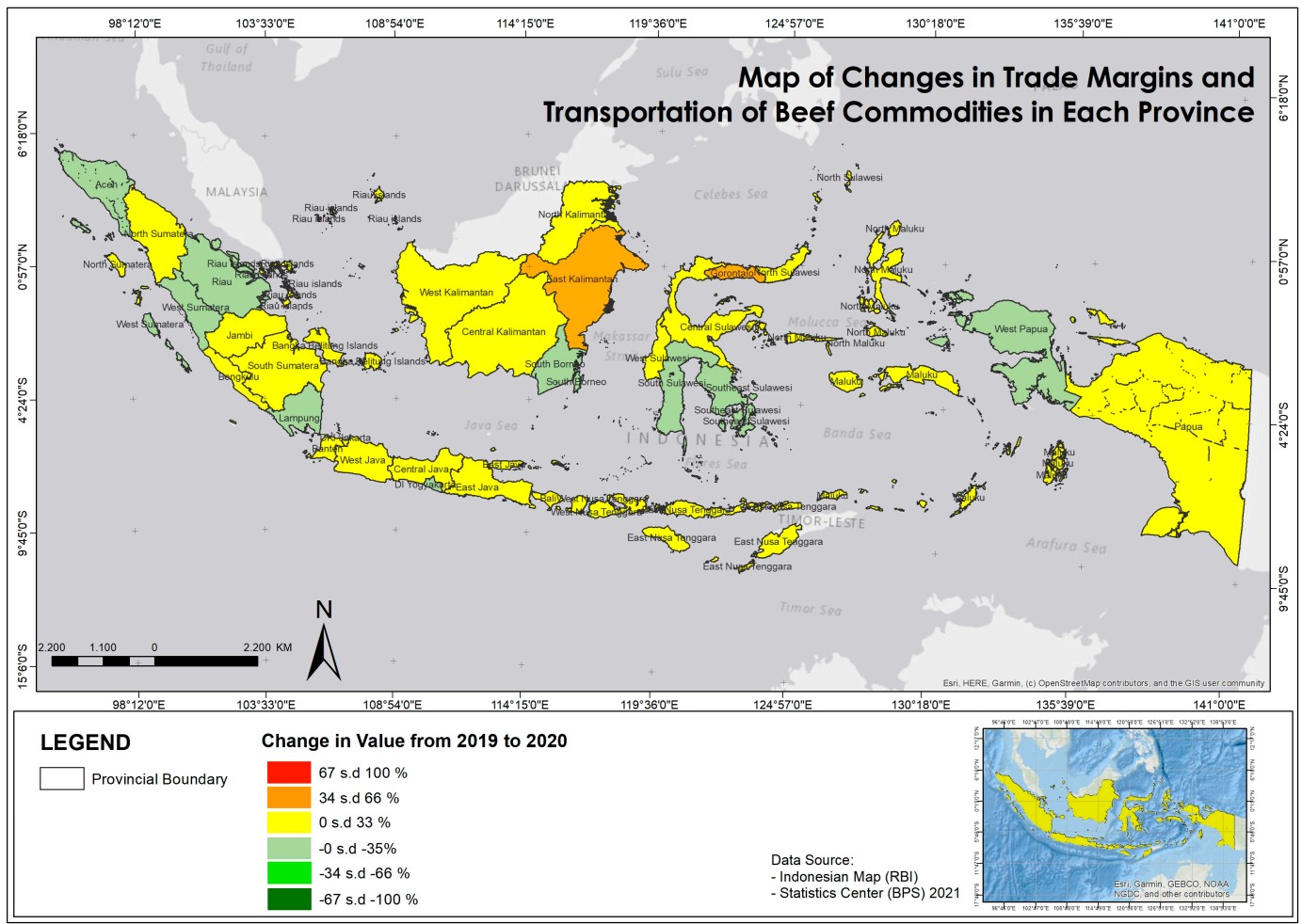

**Figure 4.** Map of changes in the beef commodity marketing margin at the provincial level (source: Indonesian Map and Statistics Center 2021, analysed).

*3.5. The Margin Change Pattern for the Chicken Meat Market*

The level of consumption of animal food products and their contribution to the market increases with increasing income. In low-income communities, the level of consumption of broiler meat and its contribution to the market is much higher than other animal food products. The marketing chain of purebred chicken was relatively less affected by the COVID-19 pandemic as shown by Figure 5 below. The impact of the pandemic on changes in income, only reduced consumer prices for eggs and chicken meat in Indonesia but not producer prices. Mobility restriction policies on the one hand, and the necessity of food as one of the most basic human needs on the other, triggered the development of e-commerce. Broiler meat products are also relatively easy to obtain with the many itinerant chicken traders in Indonesia. The affected areas only occur in the provinces of West Java, Banten, Central Kalimantan, North Kalimantan, South Kalimantan, West Sumatra, Aceh, and Maluku.

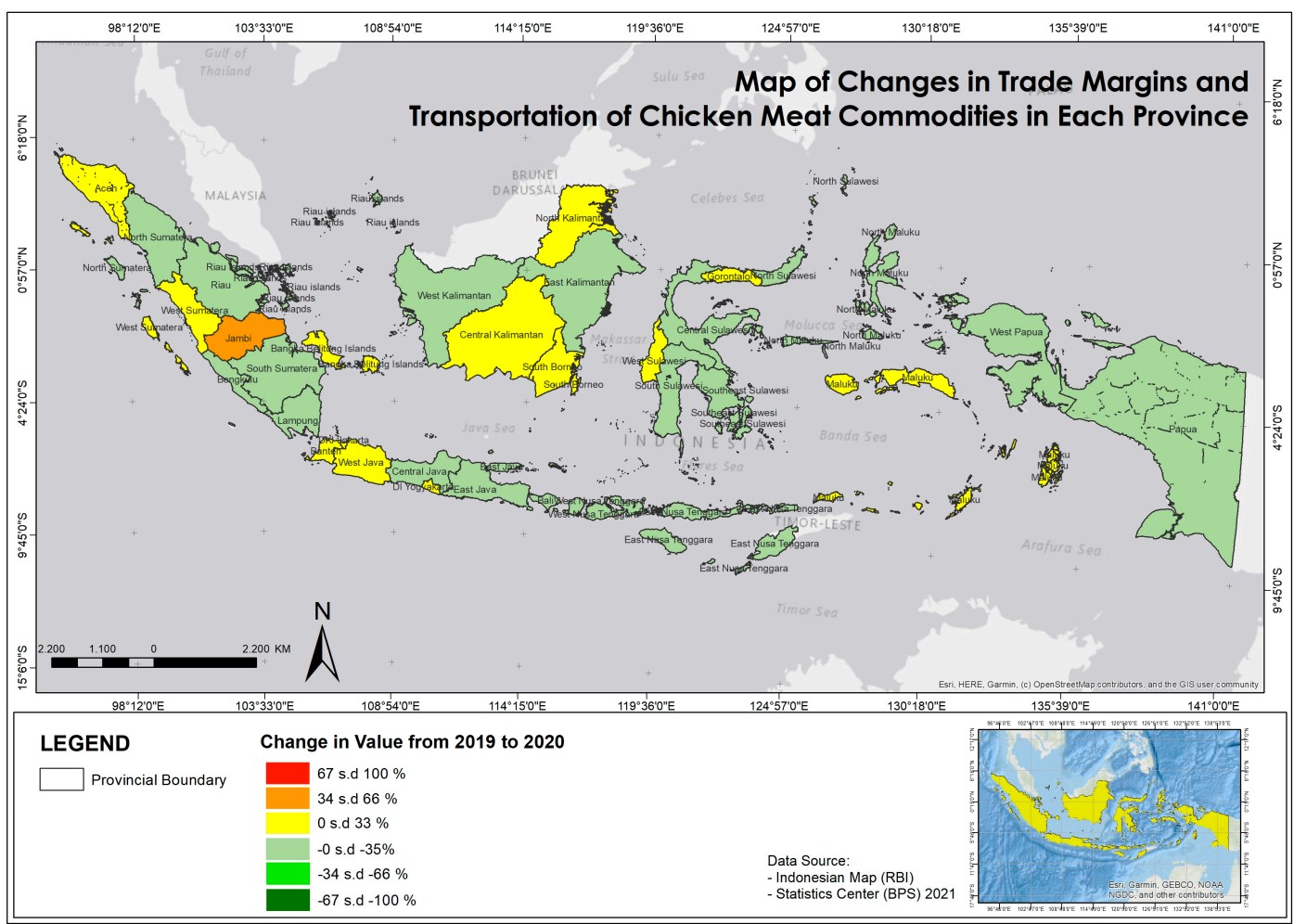

**Figure 5.** Map of changes in marketing margins for broiler chicken at the provincial level (source: Indonesian Map and Statistics Center 2021, analysed).

### 3.6. *The Margin Change Pattern for the Chicken Egg Market*

Chicken eggs are the most affordable source of protein for people in Indonesia. From the production aspect, the supply of chicken eggs was not much affected by the COVID-19 pandemic compared with normal conditions, as Figure 6 shows. Merchant turnover during the pandemic in general experienced a decline due to a general decline in income. At the beginning of the pandemic, most people tended to curb consumption. Marketing margins were relatively stable for areas in Sumatra Island, Kalimantan, West Java Island, and Central Sulawesi Island. Areas with high per capita consumption of chicken eggs experienced an increase in marketing margins, such as the provinces of South Sulawesi, Central Java, East Java, and Bali-Nusa Tenggara.

In terms of environmental conditions, chicken eggs have advantages over other sources of animal protein. Chicken eggs can be stored at room temperature for about 14 days without using a refrigerator (refrigerator or freezer). This is one of the factors driving the decision of many community members to meet their animal protein needs by increasing egg consumption, in addition to the affordable price of eggs.

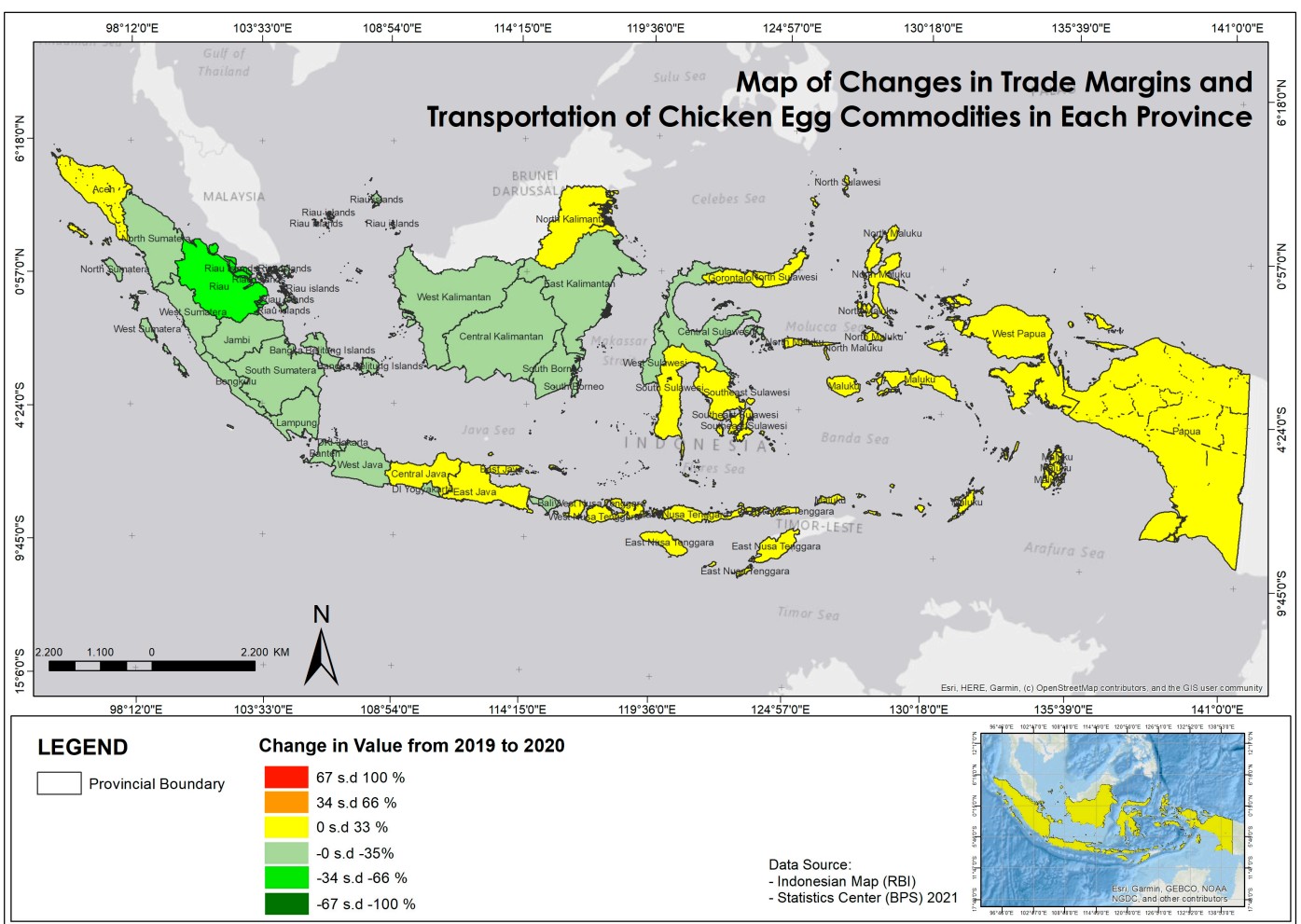

**Figure 6.** Map of margin changes in the marketing of chicken eggs at the provincial level (source: Indonesian Map and Statistics Center 2021, analysed).

### 3.7. The Margin Change Pattern for the Sugar Market

Sugar is one of the commodities with the most fluctuation in terms of demand compared with other commodities. The price of sugar at the retail level skyrocketed in the midst of the COVID-19 pandemic. This occurred after the government implemented mobility restriction policies in a number of areas, especially the urban areas of Jakarta and its surroundings. The increasing transmission of COVID-19 limited the country's full reopening after restrictions were imposed in March 2021, and the result was that many customers avoided restaurants and food stalls. Meanwhile, sugar prices remained high in most parts of the archipelago because domestic logistical disruptions caused an uneven supply of sugar. Areas that experienced disruption in sugar distribution are those that exhibited an increase in marketing margins, including West Java and Central Java, the eastern region of Sumatra Island, North and West Kalimantan, most of the islands of Sulawesi, Nusa Tenggara, and Papua. The effects are shown in Figure 7 below.

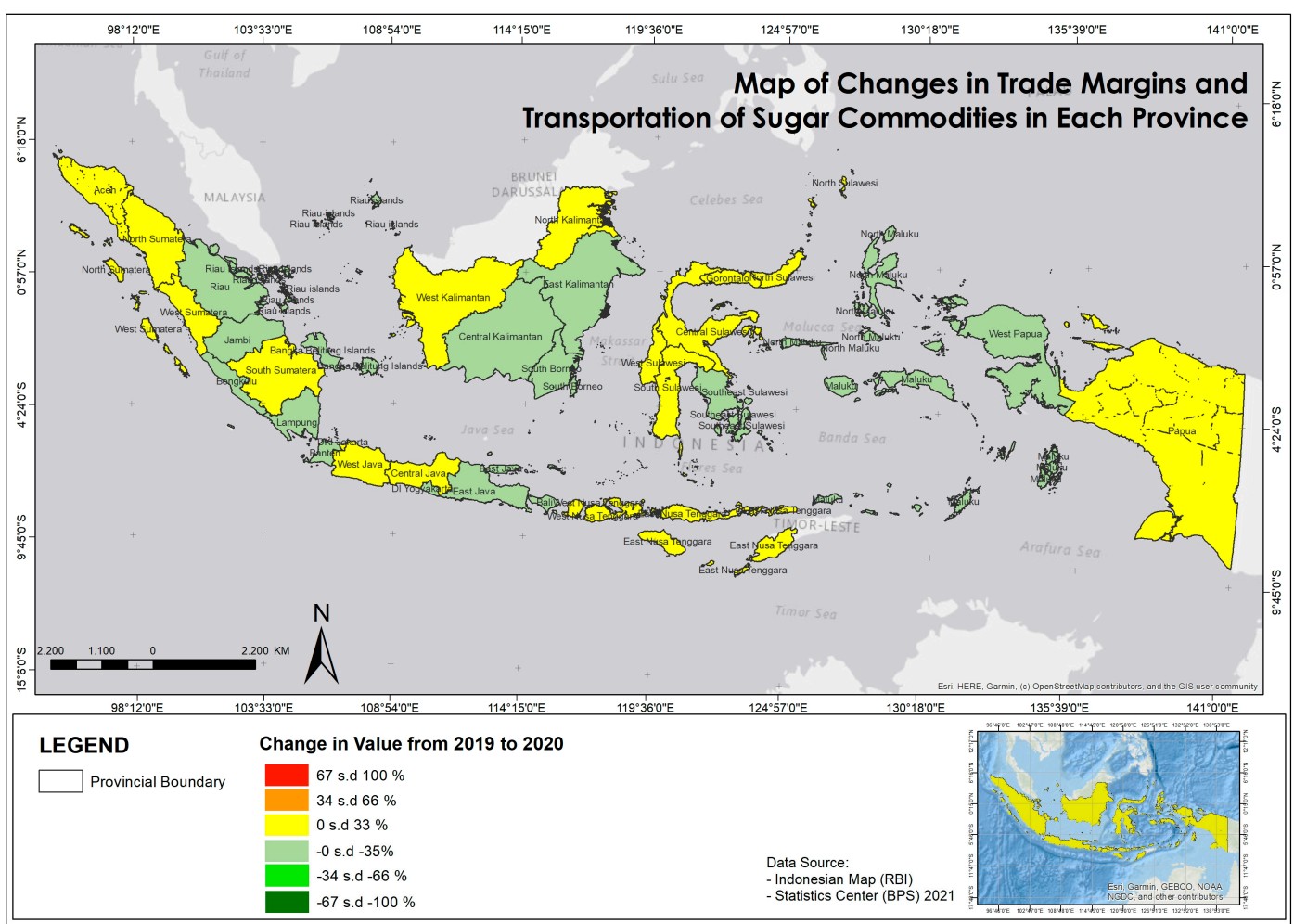

**Figure 7.** Map of margin changes in the sugar commodity market at the provincial level (source: Indonesian Map and Statistics Center 2021, analysed).

### 3.8. The Margin Change Pattern for Cooking Oil

Cooking oil is still one of the commodities with the greatest impact on inflation, at 0.01 percent in January 2022 (BPS 2023). There are many products derived from cooking oil in the processing industry, and many small and medium-sized businesses depend on cooking oil for their production inputs. Based on the data on changes in trading margins, the price of cooking oil has slightly increased when the average price before COVID-19 is compared with the average price after COVID-19. The price of cooking oil during the pandemic in Java was relatively good, except for Central Java. For Sumatra Island, only the provinces of South Sumatra and Jambi experienced an increase in trade margins. For Kalimantan Island, Central Kalimantan and East Kalimantan increased in trade margins. Furthermore, the provinces that experienced an increase were the provinces of South Sulawesi, West Sulawesi, North Sulawesi, North Maluku, West Papua, and Nusa Tenggara.

Distribution channel limitations due to mobility restrictions are believed to be the main factor underlying the increase in trading margins shown in Figure 8 below. The pandemic has disrupted the world's supply chains, and the demand remains high. Meanwhile, limited supplies have caused an increase in CPO prices since 2020. The anomaly is that this high price is due to high demand as a result of disruption in the world's oil supply. Meanwhile, not all of the domestically produced CPO is used for the integrity of cooking oil. There are at least 120 derivative products produced from processed palm oil. In addition, the need for exports has increased, leading to high demand and prices.

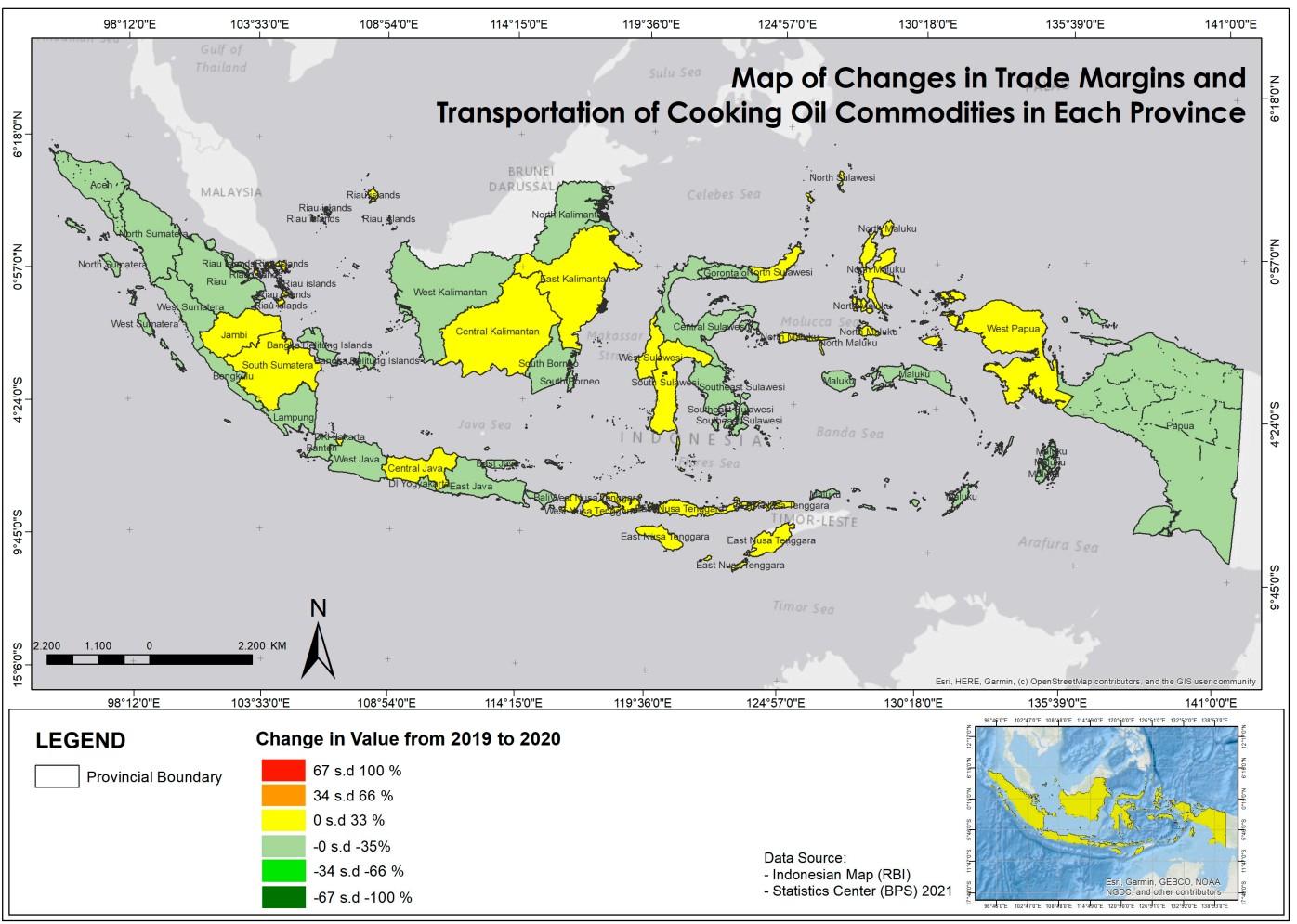

**Figure 8.** Map of changes in the marketing margin of cooking oil commodities at the provincial level (source: Indonesian Map and Statistics Center 2021, analysed).

## 4. Discussion

The supply chain of an agricultural commodity includes supply chain management, which involves the management of the flow of good, the flow of information, safe transaction methods for the flow of mone and effective and efficient processes for procurement, storage, transportation, distribution, and delivery services in terms of type, quality, quantity, time, and the desired location (Min et al. 2020).

Most of the studies that have been published on the impact of COVID-19 on agribusiness supply chains have focused on developed countries. They have reached a broadly agreed set of conclusions. The first stage of the pandemic was an initial panic: wholesale markets diminished when restaurants, hotels, and schools were closed. With more food consumed at home, grocery retail demand for flour and other baking products increased significantly, which threatened food availability even in agriculturally rich countries such as the USA (Anderson et al. 2021) and New Zealand (Hall et al. 2021). Just-in-time food supply chains, with package sizes and infrastructure intended for wholesale buyers, struggled to adapt to retail sales.

Soon, even in countries such as Australia and Canada with well-managed supply chains, employee shortages as a result of both travel restrictions and illness emerged in the agriculture sector, and consumer confidence was shaken (Jones et al. 2022), with effects extending from large, concentrated meat and dairy processors to transportation networks (Hobbs 2020, 2021). Some products such as milk, vegetables, and even livestock were dumped as a result of slow responses to consumer choices (Yaffe-Bellany and Corkery

2020). In short, the pandemic exposed the 'brittle' (Weersink et al. 2021, p. 2) or 'fragile' (Marusak et al. 2021) nature of just-in-time supply international agribusiness chains. The result was that the short-term wholesale prices doubled due to supply constraints, although consumers suffered less, with official sources indicating that retail prices also increased on average by only 10% in Canada and 30% in the USA (Weersink et al. 2021, p. 3). Undoubtedly, price volatility for staple commodities increased significantly as a direct result of the pandemic (Laili et al. 2022).

Worldwide, the downstream stages of the supply chain experienced larger disruptions than the production stage (Mogues 2020, p. 3). There were some differences observed both internationally and between sectors. On the whole, in developed countries, the pandemic had a great impact on perishable agricultural products, such as vegetables, fruits, and especially livestock and poultry (Hayes et al. 2021; Weersink et al. 2020). Meat processing firms in particular experienced extreme production risk during the first stages of the pandemic (Virmond et al. 2021, p. 742). By contrast, scholars' ability to forecast the relatively modest effects of the pandemic on grain and oilseed production, with no real threat to the provision of staple foods in Canada (Brewin 2020, 2021), on international trade for staples beyond initial export restrictions (Barichello 2020), and on the bulk grain transportation (Gray 2020) was both consistent and impressive. Markets concurred: Vercammen (2020) noted that even early on during the pandemic, the prices in the market tended towards eventual abatement due to the short-run surge in flour and longer-term income concerns, combined with the negative impact of WFH on grains used for ethanol.

In the second stage, however, agribusiness moved back to relatively normal conditions very rapidly, with prices and production levels rapidly returning to levels similar to those typically observed in years prior to the pandemic worldwide, including in complex markets like China, where the organisation of online markets by the government played a key role in managing supply and demand (Pu et al. 2021). Absenteeism remained a problem (Weersink et al. 2021, p. 2), and there were sporadic shortages of specific items in supermarkets, but there was no further major panic. Even the effect on beef and sheep farming turned out to be relatively small, short-lived, and largely offset by other global influences, although in middle-income countries such as Brazil and Argentina, there were shifts in consumer demand, which was reduced for beef and sheep meat in favour of lower-priced meat such as pork and poultry (Almadani et al. 2021). It was suggested that the specialisation and fragmentation of developed country 'just-in-time' agribusiness supply chains, which had led to the initial disruptions, may have also been responsible for its rapid rebound (Weersink et al. 2021, p. 15).

Any further long-term consequences, such as negative consequences for the labour markets as a result of a spur in the adoption of automation or further consolidation of supply chains, remain just speculation, and they are so far unsupported by data, and depend on changes in post-COVID-19 behaviour (Weersink et al. 2021, p. 13). Even increased online food shopping caused by the pandemic served only to accelerate the decline in high-end street restaurants that were already well underway before the pandemic (Weersink et al. 2021).

Globally, perishable agricultural products, such as horticultural products and vegetables, experienced a decrease in prices at the farm level and an increase in prices at the retail level, i.e., a widening of the marketing margin (Rifin 2022, p. 92), as evidenced, for example, in China (Zhou et al. 2020) and the United States (Nzeyimana et al. 2022, p. 4). To date, as far as developing countries are concerned, research has been more limited. There is widespread agreement that COVID-19 brought economic activity to a temporary virtual standstill and hindered agricultural production across the developing world, compounding existing problems such as climate change and regulations not supporting agriculture (Mukiibi 2020, p. 627) and hampering efforts to improve national logistics systems. Problems arose from disruptions imposed on suppliers, producers, retailers, wholesalers, and all links in the supply chain (Cardoso et al. 2021), with the disruption of the food distribution

chain and reduced community food security (Nchanji et al. 2020; Huang 2020; Pan et al. 2020; Istianingsih 2021).

The UN Food and Agriculture Organisation (FAO) quickly reported on the unfolding crisis: In Somalia, Afghanistan, and East Africa, COVID-19 caused an acute food crisis and insecurity, whilst in Bangladesh, it led to the disruption of transportation and falling prices for food products, resulting in trade barriers for food-importing countries and regions such as the Caribbean, Ecuador, and Venezuela (FAO 2020). Academic analysis was also prompt: in Malaysia, the inability of local open markets to operate within social distancing restrictions led to them being closed down. In a parallel process to that occurred in developed countries, this led to farmers giving away or dumping their perishable farm produce (Ng and Wahid 2020). With foreign workers, refugees, and those from lower-income groups facing difficulties in accessing adequate food, both government and volunteers sought to provide emergency supplies (Chin 2020, p. 162). Similarly, in India, for example, the government supported its farmers with compensation for lost crops and by purchasing unsold agricultural produce, although social distancing also proved problematic for Indian farmers (Mukhopadhyay 2020). In China, a survey of Chinese farmers and enterprises conducted in February 2020 found that 60 percent of the farmers encountered a shortage of inputs, with a lack of feed leading some farm animals to starve to death (Zhang et al. 2020). Developing countries were not immune to labour supply problems either, which were associated with workers off sick and border closures (Stephens et al. 2020). However, despite individual contributions of this kind, the impact on agribusiness supply chains in developing countries remains generally under-researched.

Even though Indonesia did not enact strict lockdown measures, some regions did come close to the implementation of Pembatasan Sosial Berskala Besar (PSBB: Large-Scale Social Restrictions) or limited certain activities. In response to the heavy impact on food security, the Indonesian government implemented several strategies, such as controlling food prices, which led to relatively stable and predictable prices (Mardianto et al. 2020); expanding agricultural land areas (Darma and Darma 2020, p. 377); regulating food distribution during PSBB; increasing food-waste awareness; compensating or providing subsidies for farmers through direct cash transfer; using the Village Funds Program and other similar assistance policies (Ulfa et al. 2021) buying unsold agricultural products; minimising unnecessary food imports; optimising the BULOG's role in releasing food stock; and increasing dietary awareness to increase immunity (Rozaki 2020, p. 254; Rozaki 2021). In terms of prices, significant variations were quickly observed between regions (World Food Programme 2020).

The empirical analysis of individual agribusiness sectoral responses to COVID-19 in the Indonesian context has revealed a perfect storm of problems. On the one hand, a study of farmers in the Tanah Datar District indicated that almost half had difficulty accessing agricultural inputs such as seeds, fertilisers, and pesticides during the pandemic (Triana et al. 2021). Another study indicated an even higher number—over four-fifths (Ulfa et al. 2021). On the other hand, studies such as those by Nurahmi and Zalizar (2021) and Surni et al. (2020) indicated that restrictions imposed as a result of COVID-19 cut demand for broilers in regions such as Jombang: To avoid a vicious spiral of reduced production and increased production costs, breeders sold chicken meat, including online, at a slightly lower than normal price in the middle part of 2020. Chicken farmers suffered losses because hotels, restaurants, and catering businesses that purchased chicken products were not operating (Yunita and Hasibuan 2021). Similar results emerged from a study (Azhari 2021) of reef fishers in the Spelman Strait: They were forced to sell their catch at low prices. Some fish were ultimately distributed to families or were processed or stored as salted fish. The reality for Indonesian agribusinesses during the pandemic was evidently not as satisfactory as for developed economies, although it mirrored the experiences of other developing countries described above.

The COVID-19 pandemic has had a significant impact on global supply chains, leading to disruptions and challenges for businesses across various sectors. The pandemic has

created various challenges for supply chains, including demand fluctuations, transportation disruptions, inventory shortages, and workforce issues (Raj et al. 2022). These challenges have highlighted the need for more resilient and flexible supply chain strategies. The supply chain challenges arising from the pandemic are interconnected, with disruptions in one area often leading to ripple effects in other areas. For example, transportation disruptions can lead to inventory shortages, which in turn can affect production and customer satisfaction (Raj et al. 2022). To address supply chain challenges, companies have implemented various mitigation strategies, such as diversifying suppliers, increasing inventory levels, and adopting digital technologies for better visibility and collaboration (Raj et al. 2022). These strategies aim to enhance the resilience and agility of supply chains in the face of future disruptions. The pandemic has had both positive and negative impacts on supply chains, depending on the industry and context (Moosavi et al. 2022).

While some sectors have experienced severe disruptions, others have found new opportunities for growth and innovation (Heidary 2022). Life sciences companies, for example, reported few effects from the pandemic (Heidary 2022). The pandemic has accelerated many pre-existing trends in supply chain management, such as digital transformation and automation. Companies are increasingly investing in technology to reduce employee exposure to COVID-19 and improve operational efficiency. The COVID-19 pandemic has served as a learning experience for supply chain management, highlighting the importance of risk and resilience planning. Companies are now reimagining their supply chain strategies and finding ways to include operational excellence and standard work to enable continual supply chain cost reduction. While there have been numerous studies on the impact of COVID-19 on supply chains, few stand out for providing a roadmap on how to recover from the pandemic consequences (Younis et al. 2023). Future research should focus on understanding the long-term effects of the pandemic on supply chains and developing strategies for building more resilient and sustainable systems (Younis et al. 2023).

## 5. Conclusions

In general, during the pandemic, food prices throughout Indonesia showed no consistent pattern; within the general framework of relative predictability, there were varying increases and decreases comparing the average price in the previous period and the average price during the COVID-19 pandemic. The explanation lies in the fact that there were two conflicting trends: weakening food demand at the consumer level versus bottlenecks in the main food supply chain. What did however become temporarily clear was that again in general, the pandemic in Indonesia resulted in increased variability of marketing margins. This was influenced by the distance from consumers to producers and the logistical complexity of transportation modes in an area. However, the marketing margin in the market for perishable foodstuffs (shallots and red chillies) was relatively larger than the marketing margin for non-perishable foodstuffs (rice, cooking oil, chicken meat, beef, chicken egg, and granulated sugar).

Although these margins have now broadly returned to pre-pandemic levels (BPS 2023), the policy implication of this analysis is that the control of prices overall may be only one aspect of overall price management during a crisis. If the objective is to ensure the smooth continuation of the supply chain, then it may be preferable to exchange some degree of price stability for a similar marketing margin stability. Alternatively, if prices cannot be allowed to take the strain, more direct methods of support might be considered for those marketing chains that are adversely affected.

The extent to which this policy conclusion can be generalised beyond Indonesia, however, depends on the relatively precise replication of this analysis in other jurisdictions, which is the call to action that this research indicates is now required. To properly maintain logistics and supply systems, the government and stakeholders must work together. This can encourage investment in transportation infrastructure, technology, and other areas to improve efficiency. To diversify the market, business stakeholders in the province should look for new opportunities. The use of digital solutions in trade and distribution

can improve crisis preparedness. Developing strategies to improve provinces' economic performance can benefit from economic sector diversification, investment promotion, and local community-based economic development. Interprovincial cooperation can increase effectiveness and promote products and services that are truly beneficial. Strategies should be developed to prepare for potential future crises by emphasising emergency plans, workplace training, and improving the capacity of safety and security systems.

The final observation, however, must be a salutary one indicating that the trajectories of marketing margins during and after the pandemic are one example of a wider perspective in relation to the effect of the pandemic. Early on, it was believed that COVID-19 might provide an opportunity to rethink a range of existing policies and practices concerning the food supply chain (Chin et al. 2020). It would, even in developing countries, drive the use of technology such as automation and robotics (Galanakis et al. 2021, p. 197), the IoT (Dutta and Mitra 2021), or drones (Kim et al. 2021). It would also help reshape existing food systems in the direction of local and regional food systems (Thilmany et al. 2021) and transition to new kinds of systems that support rural development and healthy, more plant-based diets for all members of society (Aday and Aday 2020, p. 5). It would also increase pressure to protect the environment and better align food production and consumption to the principles of sustainable development (Mollenkopf et al. 2021). With the benefit of even three years of hindsight, it has become evident that these expectations were misguided: the pandemic was transitory rather than transformative and has in fact had relatively little permanent effect.

The COVID-19 pandemic offers unusual resistance to supply chains, and the current study explores the complexity of such dynamics in the context of the pentahelix. Governments must play a key role in adapting national supply chains and ensuring their long-term sustainability. The goal of businesses is to continuously innovate and adapt to reduce risk and improve operational efficiency. Academia has an important role in providing knowledge and resources to support the planning and development of long-term solutions. Society must participate in changing consumer preferences and driving economic growth. The ability to convey accurate information and maintain public opinion is the power of the media. Flexible and tangible supply chains can be created to ensure resilience in the face of future challenges through joint work and active collaboration among the five stakeholders.

**Author Contributions:** Conceptualisation, E.P.Y. and J.R.; methodology, E.P.Y. and J.R.; software, E.P.Y.; validation, J.R.; formal analysis, E.P.Y. and J.R.; investigation, E.P.Y. and J.R.; resources, E.P.Y.; data curation, E.P.Y.; writing—original draft preparation, E.P.Y.; writing—review and editing, J.R.; visualisation, E.P.Y.; supervision, J.R.; project administration, E.P.Y.; funding acquisition, E.P.Y. All authors have read and agreed to the published version of the manuscript.

**Funding:** This research was funded by Universitas Padjadjaran, grant number 1427/UN6.3.1/LT/2020 and the APC was also funded by Universitas Padjadjaran.

**Institutional Review Board Statement:** Not applicable.

**Informed Consent Statement:** Not applicable.

**Data Availability Statement:** The datasets used and/or analyzed during the present study are available from the corresponding author upon a reasonable request.

**Acknowledgments:** We would like to express our gratitude to Universitas Padjadjaran for providing research funding, and also for providing funding for publication.

**Conflicts of Interest:** The authors declare no conflict of interest.

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
