# Peer review of "How Was the Staple Food Supply Chain in Indonesia Affected by COVID-19?"

_economies, doi:10.3390/economies11120292_

Round 1

Reviewer 1 Report (Previous Reviewer 1)

Comments and Suggestions for Authors

The author (s) developed the manuscript according to the suggestions, and they conducted the manuscript more transparently and of higher value in terms of scientific content. 

I detected moderate points (mistakes) to edit where changes are necessary before publishing: 

1. It is not recommended to start the introduction with the "main goal" without the problem statement. Remove this sentence from the first paragraph, please. (The last sentence of this paragraph repeats this goal as a research question.) 

2. It continues to be a problem that although the maps (Figure 1-8). I
didn't understand what kind of data was used. What was the source of these data? What kind of analysis is behind it? What was the software?  I miss the references for these figures in the text, insert them. 

Comments on the Quality of English Language

A thorough grammar and spelling review is required, e.g. 4 lines significant without "i", staple without double "p".

Author Response

Dear Reviewer 1,

We have attached some improvements in accordance with the direction of the comments you have provided regarding the additions:
1. introduction section; and
2. explanation of what data is used in the discussion.

Best Regard,
Eka

Reviewer 2 Report (Previous Reviewer 3)

Comments and Suggestions for Authors

Dear Authors,

I appreciate your efforts in improving the article and including my content-related remarks. However it still needs minor improvement.

There are many linguistic mistakes, ex. line 20 “The mail goal of the research…” etc.

I suggest to implement the “Discussion section”. In the "Discussion", the authors should compare and contrast the similarity/dissimilarity of findings in this research to the findings of other studies. This way, the authors can argue how this study stands among related studies, convincing readers of the originality or confirmatory values of this study toward the literature.

What can be the study limitations?

At the conclusion of this work, the future research directions should be mentioned.

Author Response

Dear Reviewer 1,

We have attached some improvements in accordance with the direction of the comments you have provided regarding the additions:
1. Corrections to language/word writing errors;
2. Addition of discussion after the research results;
3. Limitation of research; and
4. Conclusions for future research.

Best Regard,
Eka

Reviewer 3 Report (Previous Reviewer 4)

Comments and Suggestions for Authors

The abstract contains main required components, and it forms coherent text with logical conclusions and interactions between its immanent parts. The title of the paper is well formulated and it covers the content. The introduction logically follows the aim of the paper and it provides valuable introspection into unsolved topic. Methodological part of the paper is suits current scientific standards. The appropriate statistical apparatus is described adequately. The results are presented clearly. The interpretation of tables and figures is acceptable. The level of the author’s knowledge is satisfying. It is obvious that authors are well oriented in the topic and that they use appropriate terms. The overall level of language is appropriate.

However:

1) Authors have to provide more emphasis on the novelty of their work compared to existing ones. It is not clear enough the genuine contribution of the current work compared to extant studies in the literature.

2) Wider context of the presentation of the results should be applied to make the results really understandable for the audience. The results and discussion is largely underdeveloped. Authors, authors have to provide more discussion comparing and contrasting their results from existing studies, pointing out the novelty of their work. This section has to be deeply revised.

3) A large part of the conclusions is not appropriate, as it is only a concise repetition of the comments. In the conclusions, you should simultaneously consider all you have discovered, and exploit it to add something new (or new interpretations), and policy indications.

4) The authors do not discuss possible limitations of their study or the insights for future directions of research. Maybe they could discuss external validity of the results in terms of possible insights in other countries and/or additional variables that they would have liked to have to better answer to their research question.

The paper is suitable to be published after minor corrections according to the comments included in the review.

Comments on the Quality of English Language

The paper need of limited corrections. The authors have done a good job. 

Author Response

Dear reviewer 1, 

Please find attached some corrections in accordance with the direction of the comments you have provided related to:
1. the novelty of this paper;
2. discussion that compares and contrasts the research results with existing research;
3. Conclusion on the impact of the covid 19 pandemic on the supply chain; and
4. Study limitations of this research.

Kind regards,
Eka

Round 2

Reviewer 1 Report (Previous Reviewer 1)

Comments and Suggestions for Authors

Dear Authors, 

I appreciate your efforts to improve the quality of this manuscript. 

Together, with corrections based on all reviewers' opinions, I accept it for publication. Before that, please, delete chapter nr. 6. 

This manuscript is a resubmission of an earlier submission. The following is a list of the peer review reports and author responses from that submission.

Round 1

Reviewer 1 Report

Comments and Suggestions for Authors

I agree with the authors, that Covid-19 influenced our staple food consumption habits, but different products and product categories are affected differently. The manuscript is focusing on Indonesia which is why the chosen products are special for this region. Partly I can not understand authors why even have chosen beef? The country has to import beef and besides that consumption is not too high to be a staple food. 

The content and form seem to follow journal requirements, however, there are some things to correct. First I have to highlight the content of the abstract. We cannot read bout the method, results, and conclusions. Among keywords, it is recommended to use different words than in the title. It is not clear what is the research problem. Authors start with the research goal without introducing the situation and problem. 

Data are coming from the Indonesian Central Statistics Agency (2020), but I can not find the source among references. Which form did you get data from? What does mean sales and purchase data? What is the difference? How did you count on them? Are they also delivered from the Indonesian Central Statistics Agency? 

In table 1 (row 96) you used bawang merah. I think it is Indonesian and correctly it is shallot or garlic.  The maps are really very nice, however, I didn't understand what kind of data is used? What kind of analysis? What was the software? What is the connection of the paragraphs between rows 297 and 300 with the discussion? I can not find it. 

 I suggest going through once again the manuscript and improving the methodology part. The language has to be corrected too. 

Reviewer 2 Report

Comments and Suggestions for Authors

1. What are the research gaps and the contributions of this manuscript compared with existing articles ? The authors didnt give a specific introduction. You cant just say that there is little empirical research into how Covid-19 has impacted supply chains. More literature need to be reviewed.

2. Some grammar and spelling mistakes need to be fixed, such as that in line 36, line 63, etc.

3. The author needs to give the meaning of the letter subscripts, especially j,i.

4. The title of the manuscript is “Changes in the marketing and supply chain”, but the authors mainly focused on the changes in Trade Margins and Transportation. Why?

Reviewer 3 Report

Comments and Suggestions for Authors

Dear Authors,

The paper provides an interesting point of view of the changes in the marketing and supply chain of staple foods in Indonesia before and after the pandemic, however it needs major improvement.

Abstract:

In the abstract there is no information about the goal of the research and the methods.

There are two full stops at the end of the sentence (line 19).

I suggest to change the keywords: “supply chain” and “stapple food” into “stapple food supply chain” and add Indonesia as well.

Introduction:

In the introduction you mention about the research question, still there is no goal of your research. I suggest to add the sentence: The mail goal of the research is to answer the question in what way the agribusiness supply chain…

This manuscript lacks a "Literature Review" section. The literature review should rigorously compare and contrast relevant literature to establish a theoretical framework for the research. As for the research framework, my suggestion is to clarify how the literature was reviewed to justify the research framework according to relevant theories in the body of literature. In fact, the authors should improve the theoretical justification of the study. Which is/are the theory/ies adopted to justify the study. I suggest to provide theoretical justification leveraging on the a relevant theory in the field of food supply chain or marketing margins. This justification could guarantee that the authors have not overlooked some relevant papers in the analysis of the theoretical background.

I don’t understand the sentence: Line – 36 – Distribution The flow…

Materials and methods:

The materials and methods are not adequately described. What methods do you use?

What is “Bawang merah” – in the table 1? There is no “Bawang merah” in the attachment.

What is “chicken beef” in the attachment? Line 594.

What is BKP – in the table 1. There is no references.

Write more about the provinces in Indonesia. The research was made in all provinces? Are there no differences between the provinces from the point of view of agribusiness?

Result:

Please describe more your results.

Conclusions:

Part of your conclusions should be in the discussion part (line 365-373).

Please describe more your conclusions. The conclusions are not sufficiently supported by the results.

References:

All the references in the text should be replaced by square bracket [1], [2], etc, and the Reference [lines 402-589] should be according the numbers not alphabetically.

Reviewer 4 Report

Comments and Suggestions for Authors

- The introduction can be expanded by adding additional literature from Agricultural Economics. 

- Discussion can be expanded by adding more discussion that compare and contrast authors' findings with those available in the literature. 

- A large part of the conclusions is not appropriate, as it is only a concise repetition of the comments. In the conclusions, authors should simultaneously consider all they have discovered, and exploit it to add something new (or new interpretations), and policy indications.

- The authors do not discuss possible limitations of their study or the insights for future directions of research. Maybe they could discuss external validity of the results in terms of possible insights in other countries and/or additional variables that they would have liked to have to better answer to their research question.

- I recommend that authors review the article thoroughly and consider using a professional proofreading service to improve the style of the article. Many sentences are unclear.

- References are not formatted according to the MDPI’s guidelines.